# Spectroscopic identification of water emission from a main-belt comet

Michael S. P. Kelley[1✉], Henry H. Hsieh[2,3], Dennis Bodewits[4], Mohammad Saki[4], Geronimo L. Villanueva[5], Stefanie N. Milam[5] & Heidi B. Hammel[6]

Main-belt comets are small Solar System bodies located in the asteroid belt that repeatedly exhibit comet-like activity (that is, dust comae or tails) during their perihelion passages, strongly indicating ice sublimation[1,2]. Although the existence of main-belt comets implies the presence of extant water ice in the asteroid belt, no gas has been detected around these objects despite intense scrutiny with the world's largest telescopes[3]. Here we present James Webb Space Telescope observations that clearly show that main-belt comet 238P/Read has a coma of water vapour, but lacks a significant $CO_2$ gas coma. Our findings demonstrate that the activity of comet Read is driven by water–ice sublimation, and implies that main-belt comets are fundamentally different from the general cometary population. Whether or not comet Read experienced different formation circumstances or evolutionary history, it is unlikely to be a recent asteroid belt interloper from the outer Solar System. On the basis of these results, main-belt comets appear to represent a sample of volatile material that is currently unrepresented in observations of classical comets and the meteoritic record, making them important for understanding the early Solar System's volatile inventory and its subsequent evolution.

Comets contain many volatiles, with water, $CO_2$ and CO often being the most abundant[4]. Of the three, water and $CO_2$ are the most readily detected in near-infrared spectra[5]. James Webb Space Telescope (JWST) observations of comet Read were taken on 2022 September 8 16:30 UTC, 95 days after its 2022 perihelion and near its expected peak brightness[6]. At the time, comet Read was at a heliocentric distance of $r_h$ = 2.428 AU, a target-telescope distance of $\Delta$ = 2.086 AU, a solar phase angle (Sun–target–observer angle) of $\alpha$ = 24.3° and an orbital true anomaly of $v$ = 28.3°. Images of the comet taken with the NIRCam instrument[7] show a cometary coma and tail (Extended Data Fig. 1). A spectrum acquired with the NIRSpec instrument[8] shows scattered sunlight and thermal emission from the dust coma and cometary nucleus, and a bright 2.7 μm emission feature (Fig. 1). The shape and strength of the feature are consistent with a cometary water vapour emission model ($v_3$ band) with a production rate of $Q_{H_2O}$ = (9.9 ± 1.0) × 10²⁴ molecules s⁻¹ corresponding to 0.30 ± 0.03 kg s⁻¹; see Methods for details. The water coma is asymmetric and predominantly in the sunward direction (Extended Data Fig. 2).

In Fig. 1, we compare the JWST spectrum of comet Read with an infrared spectrum of comet 103P/Hartley 2 obtained by the Deep Impact spacecraft[9]. The spectrum of Hartley 2 shows two prominent emission features: the $v_3$ water vapour band at 2.7 μm and the $v_3$ $CO_2$ gas band at 4.3 μm. These features are typical of previously studied comets[5,10], but comet Read lacks the $CO_2$ emission band. We calculate a production rate upper limit of $Q_{CO_2}$ < 7 × 10²² molecules s⁻¹ (99.7% confidence level), equivalent to at most 5 g s⁻¹. Together, the water detection and $CO_2$ upper limit yield a coma abundance ratio $CO_2/H_2O$ < 0.7%, a factor of around ten lower than previous spectroscopic measurements of other comets at similar heliocentric distances and a factor of three lower than the lowest previous measurement overall (Fig. 2)[5].

All previous attempts to observe volatiles in main-belt comets resulted in non-detections. Some sensitive limits were based on direct observations of water vapour emission[11,12], with production rates four to eight times that of comet Read. Other estimates were based on non-detections of CN gas and an assumed $CN/H_2O$ abundance ratio similar to other comets, resulting in water production rates ranging from approximately 10²⁴ to 10²⁶ molecules s⁻¹ (ref. 3). Given our results here, with comet Read's water production rate near the middle of the previous main-belt comet studies and the indication that main-belt comets may be extremely depleted in $CO_2$, we conclude that other species may also be depleted, and therefore the water production limits derived from CN non-detections may be much higher than reported. This conclusion is in agreement with previous predictions that the $CN/H_2O$ ratio of the general comet population may not be representative of main-belt comets[3].

Insight into the mass-loss process may be gained through an estimate of the sublimating surface area. With a cometary nucleus water–ice sublimation model, we compute an active area of 0.03–0.11 km² (Methods). The active area corresponds to the cumulative area of hypothetical pure water-ice patches distributed about the surface and in contact with low-albedo material. The calculated range results from the unknown thermal properties and rotation state of the nucleus, quantified by

[1]Department of Astronomy, University of Maryland, College Park, MD, USA. [2]Planetary Science Institute, Tucson, AZ, USA. [3]Institute of Astronomy and Astrophysics, Academia Sinica, Taipei, Taiwan. [4]Department of Physics, Auburn University, Edmund C. Leach Science Center, Auburn, AL, USA. [5]Solar System Exploration Division, NASA Goddard Space Flight Center, Code 690, Greenbelt, MD, USA. [6]Association of Universities for Research in Astronomy, Washington, DC, USA. ✉e-mail: msk@astro.umd.edu

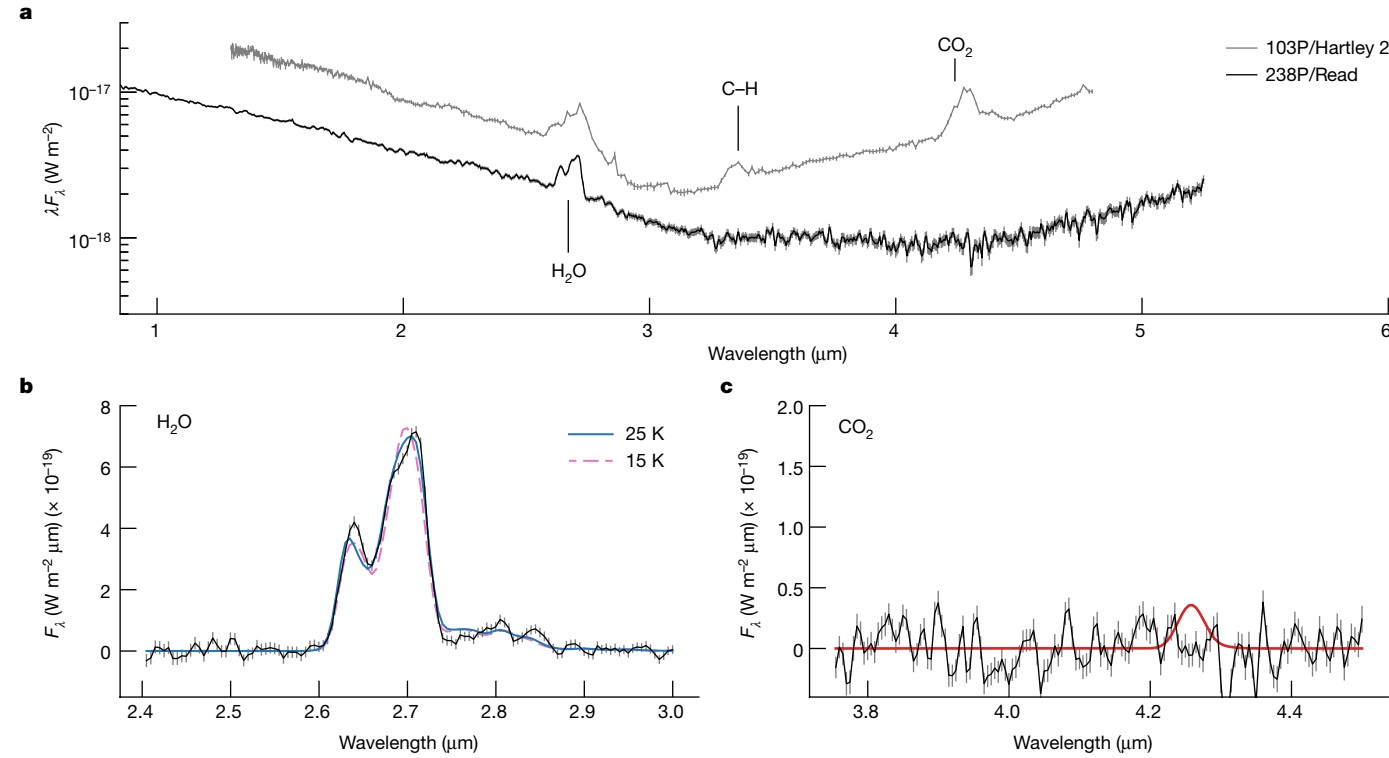

**Fig. 1 | JWST spectrum of main-belt comet 238P/Read. a**, In addition to Read, a spectrum of the Jupiter-family comet 103P/Hartley 2 from the Deep Impact spacecraft[9] is shown for comparison (scaled for display purposes). The spectral continuum varies owing to the difference in heliocentric distance of the two comets (2.4 AU for Read versus 1.1 AU for Hartley 2). Both comets exhibit a prominent water vapour emission band around 2.7 μm, but Read lacks Hartley 2's $CO_2$ emission band near 4.3 μm and the C–H stretch feature from other coma gases (approximately 3.4 μm). **b**, Continuum subtracted spectrum of the water emission band. Two best-fit water vapour fluorescence models are shown, generated with rotational temperatures of 15 K and 25 K. **c**, Continuum subtracted spectrum of the $CO_2$ emission band. A $CO_2$ fluorescence model is shown, based on our upper-limit production rate and a rotational temperature of 25 K. Error bars represent 1 s.d.

the slow rotator and rapid rotator nucleus models. The slow rotator model predicts peak water production at the subsolar point on the nuclear surface with no night-time production. The rapid rotator model would have water production equally distributed along latitudinal bands throughout the day and night hemispheres. On the basis of the observed sunward asymmetry of the water coma, we consider the slow rotator model, and therefore the lower active area, to be more appropriate. Typical comets have active fractions (the ratio of active area and surface area) less than or similar to 10% (ref. 13). With an effective radius of the nucleus, $R = 0.24 \pm 0.05$ km (ref. 14), the comet's nuclear active fraction is approximately 4–15%. Therefore, comet Read's water production rate is commensurate with its small size and the typical surface characteristics of comets.

As an alternative to sublimation distributed across the whole surface, we consider a localized source with a circular radius of approximately 100 m. Such a scenario might be generated by a small impactor that uncovered buried ice on an otherwise devolatilzed surface. Scaling previous simulations of impacts on main-belt comet nuclei[15] indicates an impactor with a diameter of around 10 m would be needed to produce a crater matching the active area. However, such an impact may be enough to catastrophically disrupt an object the size of Read's nucleus (Methods). Given our assumptions, the impactor scenario initially seems unlikely, but perhaps the parameters of subcatastrophic impacts may be tuned to produce the required active area.

In our infrared spectrum of Read, a strong, broad absorption feature is seen from approximately 2.8 to 3.7 μm. The feature is rounded with a minimum near 3.2 μm. In Fig. 3, we compare this absorption feature to those seen in comet 103P/Hartley 2 (ref. 9), 67P/Churyumov–Gerasimenko[16] and the primitive asteroid (24) Themis[17]. None are a

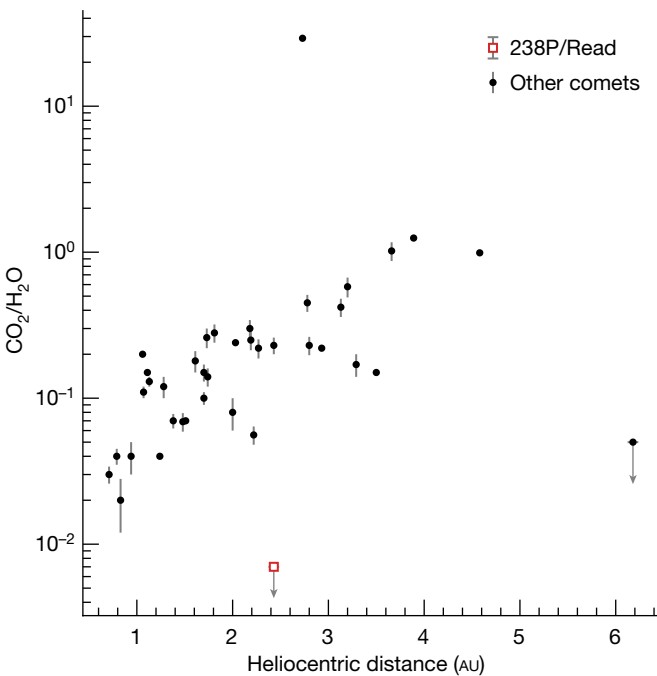

**Fig. 2 | Coma $CO_2$-to-$H_2O$ ratio of comet 238P/Read compared to the comet population.** The upper-limit coma abundance ratio (99.7% confidence) is a factor of a three lower than any previous remote spectroscopic measurement of a comet and approximately a factor of ten lower than any comet at a similar heliocentric distance[5]. Error bars represent 1 s.d.

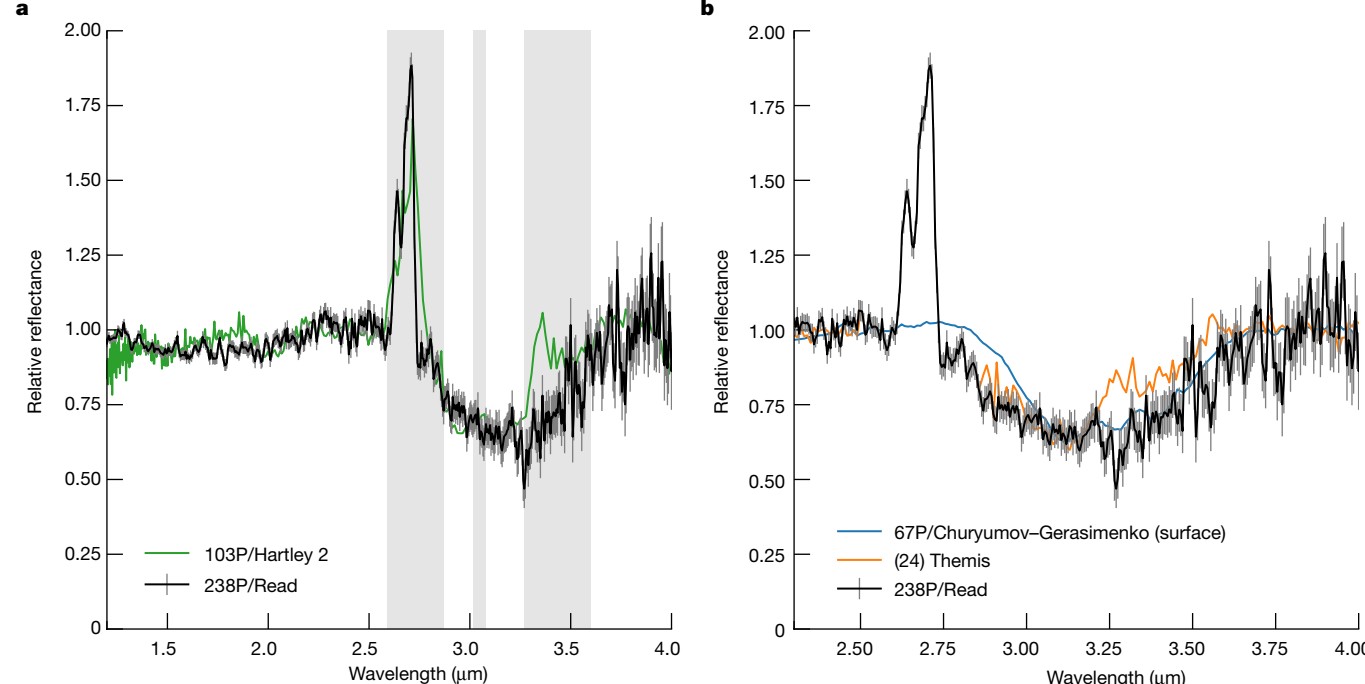

**Fig. 3 | Reflectance spectrum of comet 238P/Read near 3 μm.** The spectrum has been detrended to remove the red spectral slope. Comparison spectra have been similarly detrended and their absorption bands scaled to match the depth of the comet Read band at 3.1–3.2 μm. **a**, The spectrum is compared to the icy coma of comet 103P/Hartley 2 (band depth scaled by 0.73)[9]. Grey-shaded regions mark the presence of gas emission bands in the Hartley 2 data. **b**, The spectrum is compared with the surfaces of comet 67P/Churyumov–Gerasimenko[16] and asteroid (24) Themis[17] (band depth scaled by 2.7 and 2.9, respectively). The gap in the spectrum of Themis near 2.7 μm is due to the absorption of light by the Earth's atmosphere. Error bars represent 1 s.d.

perfect match in shape: the band of Hartley 2 is more rectangular than rounded; Churyumov–Gerasimenko matches well except for the short-wavelength edge; and Themis has a local peak near 3.25 μm that is not seen in the band of Read. Only the spectrum of comet Hartley 2 is that of a coma; the other spectra are based on observations of surfaces. Some differences may arise owing to the different scattering properties of comae grains and surfaces (comae are optically thin), but, even for a coma, particle size, shape and abundance can also play a role.

Water ice has broad absorption features at 1.5, 2.0 and 3.0 μm. These features are visible in the spectrum of Hartley 2, but comet Read's spectrum lacks any signature of water ice at 1.5 and 2.0 μm (Fig. 3). The relative strengths of the water-ice features depend on the properties of the ice, and a lack of the shorter wavelength features could be consistent with a small particle size. For Themis, radiative-transfer models indicate that a 3 μm band without corresponding short-wavelength ice absorption features can be explained by a mixture of carbonaceous (low-albedo) grains and pyroxene grains, the latter coated with a thin 10–100 nm layer of water ice[18]. However, this interpretation has since been challenged by measurements that place sensitive upper limits to water production rates for this object, ruling out surface water ice as the cause of its 3 μm band[19,20].

By contrast with the water-ice coating hypothesis, recent studies have shown that the rounded 3 μm features of large asteroids and comet Churyumov–Gerasimenko are similar to the features produced by irradiated and heated water–methanol–ammonia mixtures[21]. Separate studies of the 3 μm band of Churyumov–Gerasimenko also indicate the presence of aliphatic organics and ammonium salts[16,22]. Altogether, these results led to the conclusion that objects with rounded shaped 3 μm features may have formed at temperatures at which ammonia ice was present[21]. Furthermore, ammonia and $CO_2$ ice have similar sublimation temperatures[23], and therefore it may be that Read had both of these volatiles in the past, but they have since been lost. Additional analysis of comet Read's 3 μm feature and those of other small bodies

may provide more detailed insight into the formation or evolutionary history of (main-belt) comets and asteroids.

Dynamically, Read is closely associated with outer main-belt asteroids, as opposed to the classical comet populations such as Jupiter-family comets or long-period comets[24]. Numerical integrations indicate that even though Read's orbit has only been stable for approximately 20 Myr (compared with stability over 1 Gyr timescales for other main-belt comets[24]), it is unlikely to be a recently implanted Jupiter-family comet from the outer Solar System because of its low inclination[25]. This dynamical result is consistent with the strong depletion of $CO_2$ in the coma of comet Read reported here, which thermal modelling predicts for objects with long residence times (≥1 Myr) in the outer main asteroid belt[23].

Comet Read is also dynamically associated with an apparent cluster of low-albedo asteroids known as the Gorchakov asteroid family[26]. Asteroid family members form from catastrophic disruptions of larger parent bodies. They may have younger effective surface ages than non-family asteroids, which is thought to make the existence of near-surface ice more thermophysically plausible[26] in a region of the Solar System where ice at shallow depths is otherwise expected to be highly susceptible to depletion by solar processing[27].

The surface of comet Read appears to be devolatilizing on orbital timescales. Combining our measured dust-to-ice mass-loss rate ratio (approximately 0.3) with our measured water production rate and a few canonical assumptions, we suggest that the subsurface water-ice layer retreats faster than the surface (Methods), which should ultimately quench activity, commensurate with with previous thermophysical models[23,28]. Furthermore, this is in agreement with the observation that Read's activity appears to be declining from orbit to orbit (Methods). Together, this analysis and the decreased dust content indicate that the comet's present-day activity is a relatively recent phenomenon and not directly related to the Gorchakov family formation event. Other surface renewal processes may be needed, such as an impact by a small

asteroid[29], or surface mass loss or redistribution due to Yarkovsky–O'Keefe–Radzievskii–Paddack-effect-induced spin up[30].

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

## Methods

### Comet 238P/Read

Comet Read orbits the Sun in the outer main asteroid belt. It has a semi-major axis of 3.166 AU, a low inclination of 1.3° and a moderate eccentricity of 0.25. Perihelion occurs at a heliocentric distance $r_h$ = 2.37 AU every 5.6 years[31]. Comet Read was the second main-belt comet to be discovered and one of the three objects used to identify the population as a new class of comet[1]. It has exhibited a dust coma and tail in optical imaging observations at every perihelion since its discovery in 2005[6]. The active period ranges from 195 days before perihelion to 300 days after perihelion, with the amount of visible dust peaking approximately 100 days after perihelion[32]. The delay between the time of perihelion and the time of peak visible dust is common in the main-belt comet population[33], and in comet Read's case appears to be a consequence of a low dust expansion speed[34], which causes material to build up near the nucleus.

### Observations and data reduction

Observations of comet Read (programme ID 1252) were obtained with JWST's NIRSpec and NIRCam instruments. The JWST is a space telescope located at the Earth–Sun L2 Lagrange point with a gold-coated primary mirror and an effective aperture size of a 6.5 m diameter telescope[35]. The NIRSpec data were taken with its Integral Field Unit and prism disperser with a mid-time of 2022 September 8 16:30 UTC and total exposure time of 3,210 s. The Integral Field Unit mode slices a 3.0″ × 3.0″ field-of-view into 30 spectra, each covering a 0.1″ × 3.0″ field-of-view. The spectral wavelengths range from 0.6 to 5.2 µm, with a resolving power ($\lambda/\Delta\lambda$) that varies with wavelength, from 100 near 0.6 µm, decreasing to 30 near 1.2 µm and then increasing to 300 near 5.2 µm. The observatory tracked the comet at its predicted non-sidereal rates. Four integrations were taken with small (approximately 0.1″) movements between them to mitigate against detector artefacts and improve spatial and spectral sampling.

The uncalibrated data were downloaded from the Mikulski Archive for Space Telescopes and processed with the JWST Science Calibration Pipeline version v1.9.4 and JWST Calibration Reference Data System context file number 1041. The background was removed from the four NIRSpec exposures using observations of contemporaneously obtained blank sky, 42″ away from the comet. No sign of any signal from the comet was seen in the background data. Comet spectra were extracted from each exposure within a circular aperture radius of 0.3″, centred on the inner coma. The spectra were in agreement in regions of high signal-to-noise ratio, but the continua disagreed in regions of low signal-to-noise. The differences were mitigated with an in-scene background subtraction. The in-scene background contained little continuum, but substantial water gas emission, and therefore we based our gas band analysis on the spectra without the in-scene background subtraction. Finally, the four spectra were averaged together with outlier rejection to produce a single spectrum. The absolute calibration requirement for NIRSpec spectroscopy is 10% and we adopt this value as a minimum uncertainty for all spectroscopic results, except for those based on a relative comparison of the data (gas abundance ratios and continuum colour).

Maps of the reflected light, water emission band and continuum temperature are shown in Extended Data Fig. 2. The continuum temperature is estimated from the ratio of the mean thermal emission at 4.1 to 5.2 µm to the mean scattered light at 0.7 to 2.5 µm, assuming the scattering and emission cross-sections are equal. The calculations are based on the Planetary Spectrum Generator model dust continuum[36]. The temperature map peak is offset from the nucleus position, approximately 0.1″ north. This offset appears to be a real aspect of the data. That the nucleus itself does not stand out in this temperature map is surprising and this should be revisited as the NIRSpec spatial calibration improves with time.

JWST's NIRCam instrument captured images of comet Read immediately before the NIRSpec spectra. The camera simultaneously imaged the comet through the F200W and F277W broadband filters (24% width) using two separate detectors and a dichroic. Both detectors have dimensions of 2,040 × 2,048 pixels, and pixel scales are 0.031″ pixel⁻¹ for the short-wavelength channel and 0.063″ pixel⁻¹ for the long-wavelength channel. For a solar spectrum[37] the filters have effective wavelengths of 1.97 and 2.74 µm for F200W and F277W, respectively. Five exposures were taken with around 6″ spatial offsets between each to mitigate effects from detector artefacts, cosmic rays and background sources. The full array of all detectors were read out with the BRIGHT1 pattern, for a total exposure time of 1,020 s per filter. The NIRCam data, aligned on the comet and combined by wavelength, are shown in Extended Data Fig. 1.

NIRCam images were downloaded from the Mikulski Archive for Space Telescopes and processed with pipeline version v1.6.2 and Calibration Reference Data System context file number 969. Updated absolute photometric calibration values became available on 6 October 2022, and we scaled our NIRCam data to account for the changes. Photometry of the comet was measured within 0.3″ radius apertures: 22.84 ± 0.03 mag in F200W and 23.22 ± 0.05 mag in F277W (AB magnitude system); uncertainties are based on the standard deviation of the five exposures. These measurements include an aperture correction computed with the WebbPSF program[38] for a nominal coma surface brightness profile (−0.12 and −0.14 mag for F200W and F277W, respectively). There is excellent agreement in results from the two instruments. Synthetic photometry from the spectrum and filter throughputs yield a colour of $m$(F200W) − $m$(F277W) = −0.39 mag compared to −0.38 ± 0.05 mag from NIRCam.

### Reflectance spectrum

The reflectance spectrum is produced by dividing the NIRSpec data by a spectrum of the Sun[37]. The result shows that the coma is red coloured, with a mean (linear) spectral slope of 2.18 ± 0.02% per 100 nm between 1.0 and 2.55 µm (normalized at 2.0 µm). However, the reflectance spectrum is not linear over this wavelength range (Extended Data Fig. 3).

We assess the thermal contribution to the spectrum by assuming the scattered light has a constant spectral slope and the thermal emission can be described with a scaled Planck function. A least-squares fit to the continuum at 1.2–2.2, 2.5–2.6 and 3.5–5.2 µm ($\chi_\nu^2$ = 1.8, $\nu$ = 560) is presented in Extended Data Fig. 3. We also examined a best fit to a more limited wavelength range: 2.5–2.6 and 3.5–5.2 µm ($\chi_\nu^2$ = 1.2, $\nu$ = 360). The fits suggests the thermal emission accounts for 3 to 5% of the spectrum at 3.7 µm. The long-wavelength edge of the 3 µm absorption band is approximately 3.7 µm, and therefore thermal emission is unlikely to effect our analysis of the band shape.

### Nucleus contribution

The contribution of the nucleus to the spectrum depends on the nucleus shape and rotation state at the time of the observation, the albedo, the colour and the thermal properties of the surface. An effective nucleus radius has been measured for comet Read, assuming a spherical shape and a visual albedo of 5%: $R$ = 0.24 ± 0.05 km (ref. 14). Taking this estimate and a nominal comet nucleus thermal model[39], the nucleus model dominates the thermal emission, accounting for 98 ± 37% of the spectral flux at 5.0 µm. At 2.0 µm, reflected light from the nucleus accounts for 21 ± 8% of the spectral flux, assuming the near-infrared colour of the nucleus is similar to the colour of the coma.

### Gas coma model

A model cometary coma is used to produce a synthetic spectrum of the gas fluorescence band emission, which is compared to the data to estimate the molecular rotational temperature and production rate at the nucleus. We can use radiative-transfer models[40,41] to compute the excitation state of ro-vibrational bands of cometary gases (here $H_2O$

and $CO_2$) pumped by infrared solar radiation and collisions with other molecules and electrons.

For the coma itself, we assumed an isotropic and constant gas expansion with a speed of $v_{gas} = 850r_h^{-0.5}$ m s$^{-1}$ = 513 m s$^{-1}$ at the comet's heliocentric distance[42,43]. Photodissociation defines the lifetime and spatial extent of molecular species, but the correction of this effect is only a few per cent for our data. These assumptions are generally accurate enough (and widely used by the community) to calculate integrated column densities and molecular fluxes across the coma.

To model the gas fluorescence emission, we use the Planetary Spectrum Generator[36]. Its models incorporate excitation processes using the local thermodynamic equilibrium (LTE) and non-LTE layer-by-layer and line-by-line radiative-transfer fluorescence models using NASA-Goddard Space Flight Center, HITRAN, Gestion et Etude des Informations Spectroscopiques Atmosphériques (GEISA), NASA Jet Propulsion Laboratory and Cologne Database for Molecular Spectroscopy (CDMS) spectral databases to compute line fluxes. We assume an expanding coma, for which the fluorescence efficiencies ($g$ factors) used in synthetic emission models in this study are generated with a quantum mechanical model developed for $H_2O$ (ref. 41). This model integrates the latest radiative-transfer methods and spectroscopic parameterizations to compute high-resolution spectra by line-by-line calculations and utilizes the efficient correlated method at moderate to low resolutions.

The populations of the excited ro-vibrational levels follow a time-dependent equation[44]. At higher coma densities than comet Read, collisional excitation is the dominant process that determines rotational levels. The ground-state populations are mostly equilibrated and follow a Boltzmann distribution at the gas temperature ($T_{rot}$). In this case, the rotational temperature of different gases are usually similar. The coma is a mix of gas and dust and fully described by input parameters such as the heliocentric distance ($r_h$) and the gas production rates ($Q$). At the low gas production rates of comet Read, volume densities in the inner coma result in low molecule–molecule and molecule–electron collisional rates, and therefore do not establish the radiative equilibrium state of the molecules. Thus, the atmosphere can be considered to be in a full non-LTE state (Extended Data Fig. 4). Using the Planetary Spectrum Generator, the best fit of our models corresponds to $T_{rot} = 25$ K, considering an equilibrated rotational state and a non-LTE vibrational state in fluorescence. However, as can be seen in Fig. 1, the model is not in perfect agreement with a noticeable difference between the model and $H_2O$ feature at around 2.63 μm. The $H_2O$ spectral feature centred at 2.63 μm is better fit with $T_{rot} = 15$ K whereas the 2.69 μm feature is better fit with $T_{rot} = 25$ K (Fig. 1). In a full non-LTE regime (for example, unequilibrated rotational and vibrational states) a single temperature cannot describe the coma; therefore, this is perhaps indicative of further non-LTE effects, beyond vibrational fluorescence, or a full non-LTE state. For $CO_2$, we assumed the same $T_{rot}$ (25 K) when computing the band upper limit.

Spectra for $H_2O$ and $CO_2$ were generated for a fixed production rate using the above model. We used a least-squares method to fit the continuum (modelled as a first- or second-order polynomial) and gas emission. Uncertainties were derived using the bootstrap technique and the spectral uncertainties. This was sufficient for fitting the water band, but the $CO_2$ band upper limit required consideration of correlated noise in the spectrum. Correlated noise is typical of integral field spectrometers, and we estimated the data covariance with the Gaussian Processes technique[45] using the George Python package[46]. Uncertainties based on the five-parameter fit (production rate, two polynomial coefficients and two data correlation parameters) were derived with the Emcee Python package[47]. All four spectra were consistent with a non-detection for $CO_2$ and we report results fitting a combined spectrum. The average column density of $H_2O$ and $CO_2$ molecules within a 0.3″ radius aperture is calculated to be $2.11 \times 10^{16}$ m$^{-2}$ and less than $1 \times 10^{14}$ m$^{-2}$, respectively, and the production rates are

$Q_{H_2O} = (9.88 \pm 0.10) \times 10^{24}$ molecules s$^{-1}$ and $Q_{CO_2} < 7 \times 10^{22}$ molecules s$^{-1}$ (excluding the 10% calibration uncertainty). The $CO_2$ limit is based on the one-sided 99.7% confidence limit (approximately equivalent to a $3\sigma$ upper limit).

## Sublimation model

An ice sublimation model[48,49] may be used to better understand the mass-loss process. We use the production rate of $H_2O$ to calculate the effective active area on the surface of comet Read. Two versions of the model were used: the slow rotator model, in which every part of the surface of the comet is in instantaneous equilibrium with incident solar radiation; and the rapid rotator model, in which the nucleus rotation rate is so high that parallels of latitude become isotherms. The two models provide lower and upper limits to the inferred active area, provided that the obliquity of the rapid rotator model is 0°. For our analysis, we assume the Bond albedo for the surface to be 0.05 and the infrared emissivity to be 1. We use a sphere with a radius of 0.24 km (ref. 14) to calculate the active surface fraction. The results of our calculations are presented in Extended Data Table 1.

## Impacts and disruption

We consider whether or not an asteroidal impact could excavate a crater large enough to account for the water production rate, assuming the surface is devolatilized and the subsurface is ice rich. Previous simulations and analysis of impacts on small cometary objects show that little ejected material is re-accreted[15], and therefore we require a crater area equal to the active sublimation area. For a 10:1 ratio of crater to impactor area[15], comet Read's impactor must be around 10 m in size. Assuming a nominal impactor velocity[50] of 5 km s$^{-1}$ and 2,000 kg m$^{-3}$ bulk density, and a bulk density of 1,000 kg m$^{-3}$ for Read, the kinetic energy per target mass is approximately $2 \times 10^7$ erg g$^{-1}$. This is an order of magnitude larger than that needed to disrupt a 240 m asteroidal body[51].

## Dust-to-gas ratio

The coma dust-to-gas ratio may be measured from our data and compared to other comets. Dust mass-loss rates typically require several assumptions that together can affect the results up to the order of magnitude level, for example, dust grain density, size distribution and expansion speed. Much of the uncertainty can be addressed by fitting the morphology with a dust dynamical model. A Monte Carlo-style analysis of comet Read's 2005 active apparition with such a model found a good match to observations using a particle size distribution with a power-law index of $q = -3.5$ and grain ejection velocities of $v_{ej} = 12a^{-0.5}$ m s$^{-1}$, where $a$ is the grain radius in micrometres[34]. The estimated mass-loss rate was $dm/dt \approx 0.2$ kg s$^{-1}$ at a true anomaly $v = 31.4°$, close to the orbital position of $v = 28.3°$ at the time of the JWST observations reported here.

A less model-dependent estimate of the dust-to-gas ratio can be obtained with the cometary $Af\rho$ quantity. This parameter is intended to enable comparisons of photometric measurements of cometary comae obtained at different times and under different conditions[52]. It is given by $Af\rho = (4r_h^2\Delta^2/\rho) \times 10^{-0.4\Delta m}$, where $A$ refers to the albedo of dust grains in the coma, $f$ represents the filling factor of grains within the photometric aperture (that is, the fraction of the aperture filled by the cross-sectional area of the dust), $r_h$ is the heliocentric distance of the object in astronomical units, $\Delta$ is the telescope–comet distance in centimetres, $\rho$ is the physical radius of the photometric aperture at the distance of the comet in centimetres, $\Delta m = m_\odot - m_{com}$ is the difference between $m_\odot$, the apparent magnitude of the Sun at 1 AU in the same filter used to observe the comet (−26.64 and −26.03 mag for F200W and F277W, respectively), and $m_{com}$, the observed apparent magnitude of the comet. $Af\rho$ values are given in units of length. A dust coma in free expansion and constant dust production rate has a line-of-sight column density that scales with $\rho^{-1}$. Thus, $Af\rho$ is nominally independent of aperture size, providing a means for combining photometric data for

comets obtained at different times, by different observers and using different facilities to search for trends or make comparisons. Cometary comae are not always so idealized, and the $Af\rho$ formulation also assumes that there is no production or destruction of dust grains in the coma, so some caution must be exercised when using this parameter[53]. The original formulation of the parameter's definition also does not account for the phase angle of the object at the time of observation, but this can be remedied by applying a phase function correction to the albedo, usually denoted $A(0°)f\rho$. We assume a phase function similar to that of comet 1P/Halley[54], $\Phi(24.3°) = 0.46$.

With the NIRCam data, we compute $A(0°)f\rho = 18.7 \pm 0.5$ cm in our 0.3″ radius aperture measured at an orbital true anomaly $\nu = 28.3°$, or 15.0 cm if the around 20% nucleus contribution is removed. Using the spectrum to scale our measurement to 0.7 μm yields 11.5 cm. Compare this to the measured activity in 2005: $A(0°)f\rho = 7.86 \pm 0.39$ cm at $\nu = 31.4°$, measured in an $R$-band filter (0.64 μm) and 4″ radius aperture[34]. Read's dust tail-dominated morphology breaks the $Af\rho$-model assumption that the comet has a nominal $\rho^{-1}$ coma, and the signal-to-noise ratio of the NIRCam data does not warrant photometry measured with an aperture matching the previous ground-based data. Instead, we extrapolate the photometry from 0.3″ to 4.0″ using its measured azimuthally averaged radial surface brightness profile: proportional to $\rho^{-1.5}$ between $\rho \approx 0.1″$ and approximately 1.1″, which is in agreement with the tail-dominated morphology[55]. For a surface brightness profile following $\rho^k$, the integrated photometry scales with $\rho^{(k+1)}$ for $k \leq -1$. Altogether, the photometry scaled from 0.3″ to 4.0″ results in $A(0°)f\rho = 3$ cm. We therefore find that the activity of this comet has potentially decreased by a factor of approximately two since 2005, but this conclusion should be revisited with contemporaneously obtained optical data.

We provide two estimates of the dust-to-gas production rate ratio, both based on our measured water production rate and F200W photometry scaled to the $R$ band. The first is from our nominal 0.3″ aperture photometry: $\log_{10} A(0°)f\rho/Q_{H_2O} = -23.93 \pm 0.06$. The second estimate of the dust-to-gas ratio uses the previous dynamical analysis of the 2005 data scaled by one-half to account for the potentially lower activity level of this orbit: $Q_{dust}/Q_{H_2O} \approx 0.3$.

In Extended Data Fig. 5, we compare comet Read's $A(0°)f\rho/Q_{H_2O}$ to the general comet population, based on the survey of ref. 13 (dust values have been converted to 0° phase angle with the Schleicher–Marcus coma dust phase function[56] and OH production rates converted to water production rates following ref. 54). By this metric comet Read appears to be one of the dustiest comets, but this is probably a consequence of low dust ejection speeds. If we instead take the computed dust-to-gas mass ratio, of around 0.3, and compare it with the ratios of around 1 measured at Churyumov–Gerasimenko[57], Read appears to be instead more gas-rich relative to dust than 67P. An important caveat is that the data we are analysing span only 1 h of total observation time, and thus we lack information about the rotational context of these measurements (the comet's rotational variability and period are not known). Furthermore, there is a wide range of estimates for 67P's dust-to-gas mass ratio (see ref. 57 for a discussion and references). Therefore, our conclusions are that comet Read has a coma dust-to-gas ratio broadly consistent with the general comet population, which suggests it may have formed in a region of the protoplanetary disk with abundant water ice.

## Activity timescale

With our measured water production rate, we can estimate order of magnitude timescales for the active period of comet Read. We first neglect dust mass loss and compare the orbital water mass loss to the amount of water within a thermal skin depth. The thermal skin depth $l_s$ is computed using[58] $l_s \approx \Gamma/(c_p\rho_g)(2/\omega)^{0.5}$, where $\Gamma$ is the thermal inertia of the surface, $c_p$ is the heat capacity, $\rho_g$ is the grain density and $\omega$ is the rotation rate. With values used in the study of comet 67P/Churyumov–Gerasimenko ($\Gamma = 50$ J m$^{-2}$ K$^{-1}$ s$^{-\frac{1}{2}}$, $c_p = 500$ J kg$^{-1}$ K$^{-1}$, $\rho_g = 500$ kg m$^{-3}$)[59] and

assuming a rotation period of 5 h as an example, we calculate a thermal skin depth of 1.5 cm. Further assuming a dust-to-ice mass ratio of 1 and ice uniformly distributed over the surface, we find $3 \times 10^6$ kg of water ice within $1l_s$. With the activity model of ref. 34, $dm/dt \propto r_h^{-3}$ from −60 to +90 days from perihelion, the comet loses $3 \times 10^6$ kg of ice per orbit. This mass corresponds to 1 thermal skin depth; the depth scales linearly with the assumed dust-to-ice ratio in this approximation. Furthermore, a dust tail is observed and therefore dust is lost from the surface. Assuming the dust-to-gas mass-loss rate ratio is constant with time, and given that our estimated dust-to-gas mass loss rate ratio is less than 1.0, we suggest that the subsurface ice layer retreats faster than the surface and that the near-surface layers devolatilize on orbital timescales.

## Data availability

JWST data are publicly available from the Space Telescope Science Institute's Mikulski Archive for Space Telescopes https://mast.stsci.edu/. Reduced data used in this analysis are publicly available at Zenodo https://doi.org/10.5281/zenodo.7864044.

## Code availability

All relevant code is publicly available: the Planetary Spectrum Generator is at https://psg.gsfc.nasa.gov/; the Ice Sublimation Model at https://github.com/Small-Bodies-Node/ice-sublimation; the JWST science data calibration pipeline at https://github.com/spacetelescope/jwst; and analysis and figure scripts at Zenodo https://doi.org/10.5281/zenodo.7864044.

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

**Acknowledgements** This work benefited from support from the International Space Science Institute, Bern, Switzerland, through the hosting and provision of financial support for an international team to discuss the science of main-belt comets. We thank D. Farnocchia, M. Micheli and J. Pittichová for improving the comet's orbit, which helped make these observations a success. This work is based on observations made with the NASA/ESA/CSA James Webb Space Telescope. The data were obtained from the Mikulski Archive for Space Telescopes at the Space Telescope Science Institute, which is operated by the Association of Universities for Research in Astronomy, Inc., under NASA contract NAS 5-03127 for JWST. These observations are associated with programme no. 1252. Support for archival research programme no. 2037 was provided by NASA through a grant from the Space Telescope Science Institute, which is operated by the Association of Universities for Research in Astronomy, Inc., under NASA contract NAS 5-03127. S.N.M. and H.B.H. acknowledge support from NASA JWST Interdisciplinary Scientist grant no. 21-SMDSS21-0013.

**Author contributions** M.S.P.K., H.H.H., D.B., S.N.M. and H.B.H. conceived and designed the observational programme. M.S.P.K., H.H.H. and G.L.V. reduced the data. H.H.H. and M.S.P.K. analysed the imaging and photometry. M.S.P.K., M.S., G.L.V. and D.B. analysed the spectroscopy. All authors contributed to the interpretation of the data and writing of the manuscript.

**Competing interests** The authors declare no competing interests.

**Additional information**
**Correspondence and requests for materials** should be addressed to Michael S. P. Kelley.

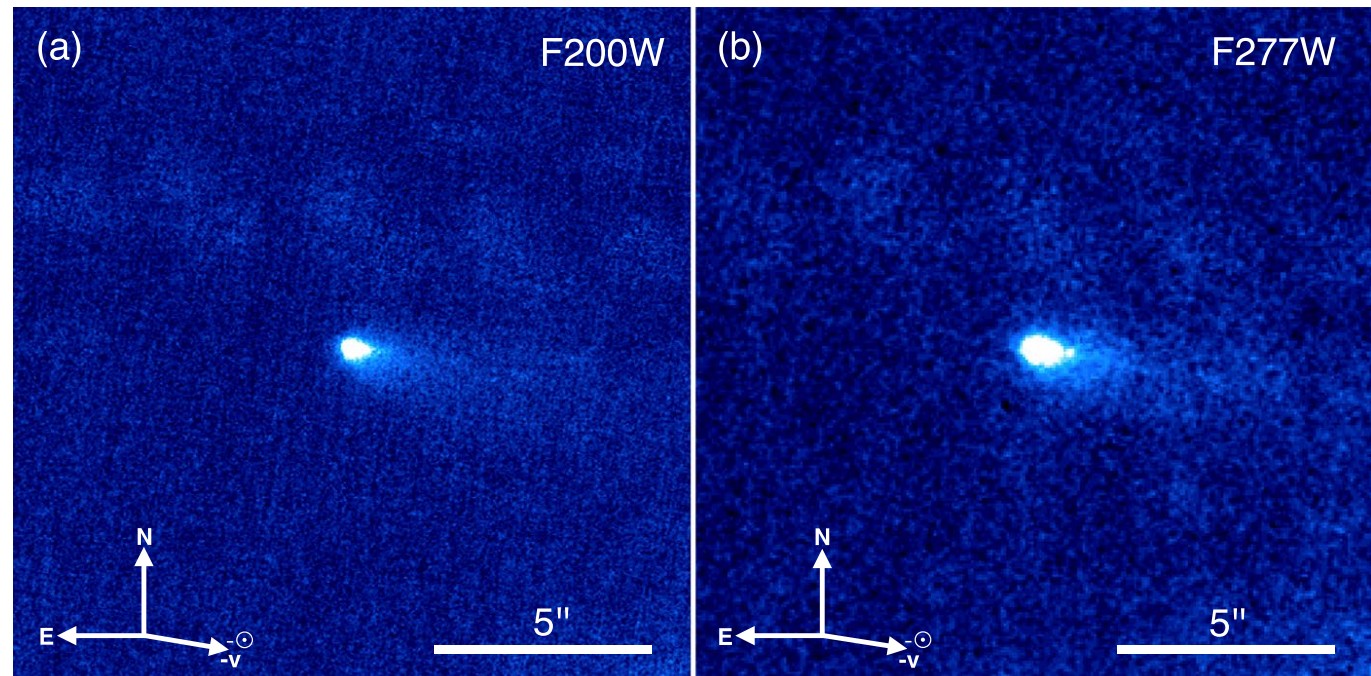

**Extended Data Fig. 1 | JWST/NIRCam images of comet 238P/Read.** Shown are images taken with the (a) F200W, and (b) F277W broadband filters. Images were combined in the rest-frame of the comet, and some artifacts are apparent from stars and galaxies moving through the background. An apparent bright spot in the F277W tail is an artifact from a single image, and does not affect our photometric results. Celestial north and east, and the projected anti-Sun ($-\odot$) and anti-velocity ($-v$) vectors are as indicated. A 5″ angular scale bar (7560 km at the distance of the comet) is also given.

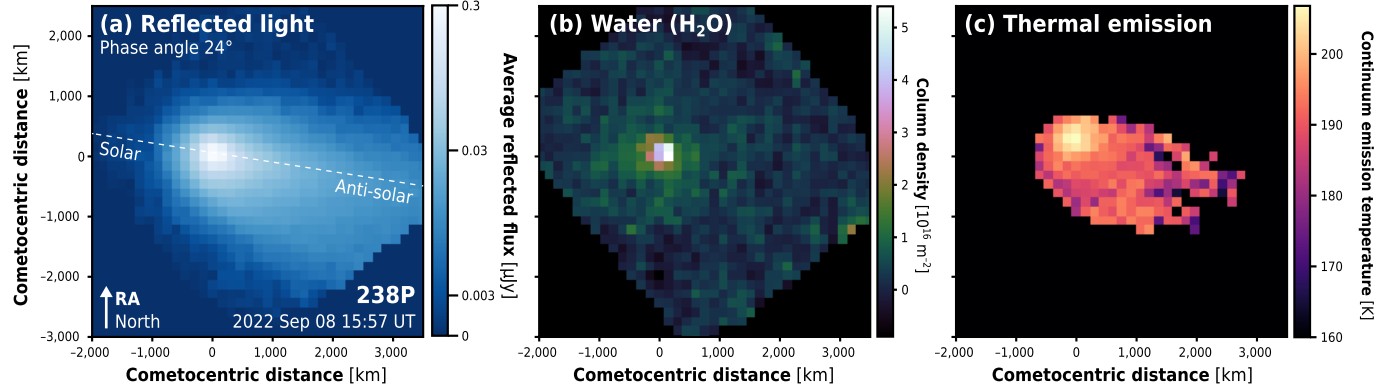

**Extended Data Fig. 2 | Comet 238P/Read dust, water, and temperature maps.** (a) Wavelength averaged spatial distribution of light scattered by dust from 0.7–2.5 μm. The brightness scale is linear from 0 to 0.003 μJy pix⁻¹, then logarithmic to 0.3 μJy pix⁻¹ (1 Jy = 10⁻²⁶ W m⁻² Hz⁻¹). (b) Water vapour column density map. (c) Approximate continuum temperature obtained by analysis of the ratio of the thermal emission at 4.1 to 5.2 μm to the scattered light map. Areas with low signal have been masked. All panels have the same orientation (Celestial north is up, east to the left), and the projected sunward and anti-sun vectors are indicated in panel (a).

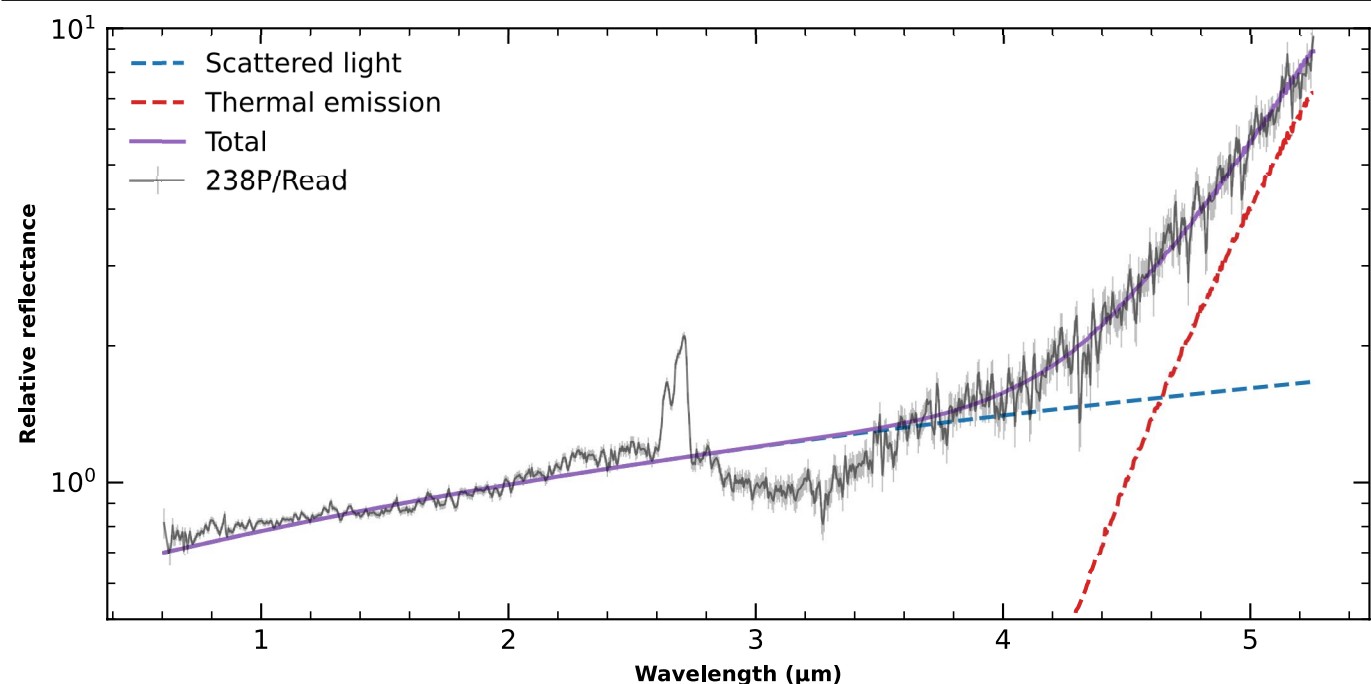

**Extended Data Fig. 3 | Relative reflectance of the coma of comet 238P/Read and best fit continuum model.** Error bars represent 1 s.d. The model assumes a constant linear spectral gradient across all wavelengths for the scattered light, and a single temperature scaled Planck function for the thermal emission.

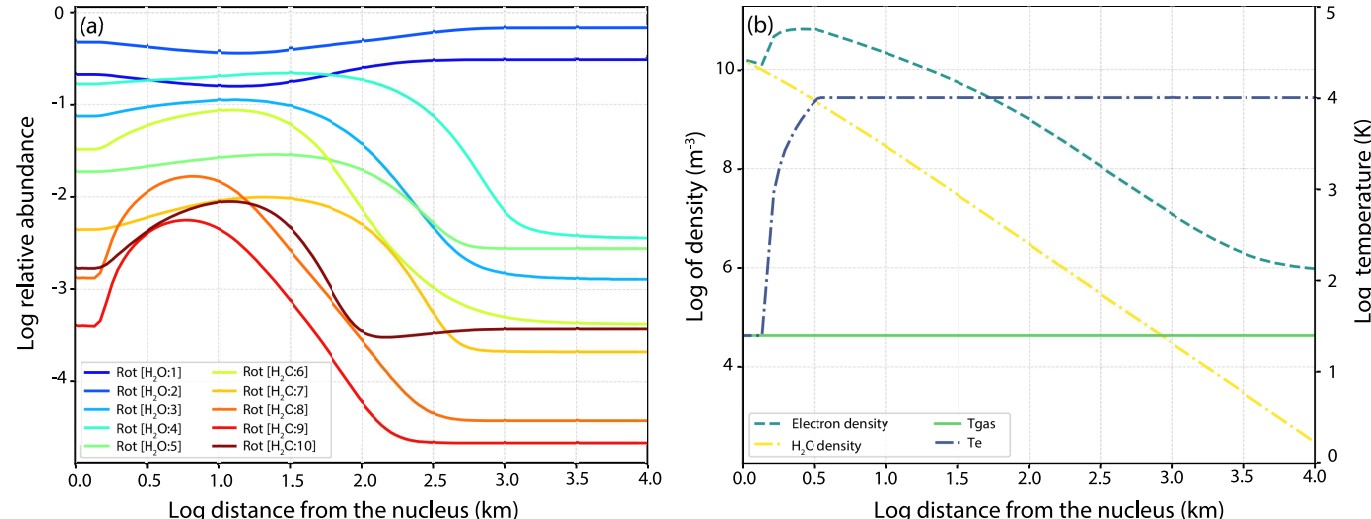

**Extended Data Fig. 4 | Water vapour rotational level populations, volume density, and temperature.** The model was computed with the Planetary Spectrum Generator[36] for $Q(H_2O) = 9.88 \times 10^{24}$ molecules s$^{-1}$ at $r_h = 2.428$ au, and $v_{gas} = 513$ m s$^{-1}$. (a) Relative population of H$_2$O rotational levels compared to all ground states including vibrational and electronic states. (b) Volume density and temperature versus distance for H$_2$O and elections. Electron collisions are negligible at these low collisional rates and were excluded.

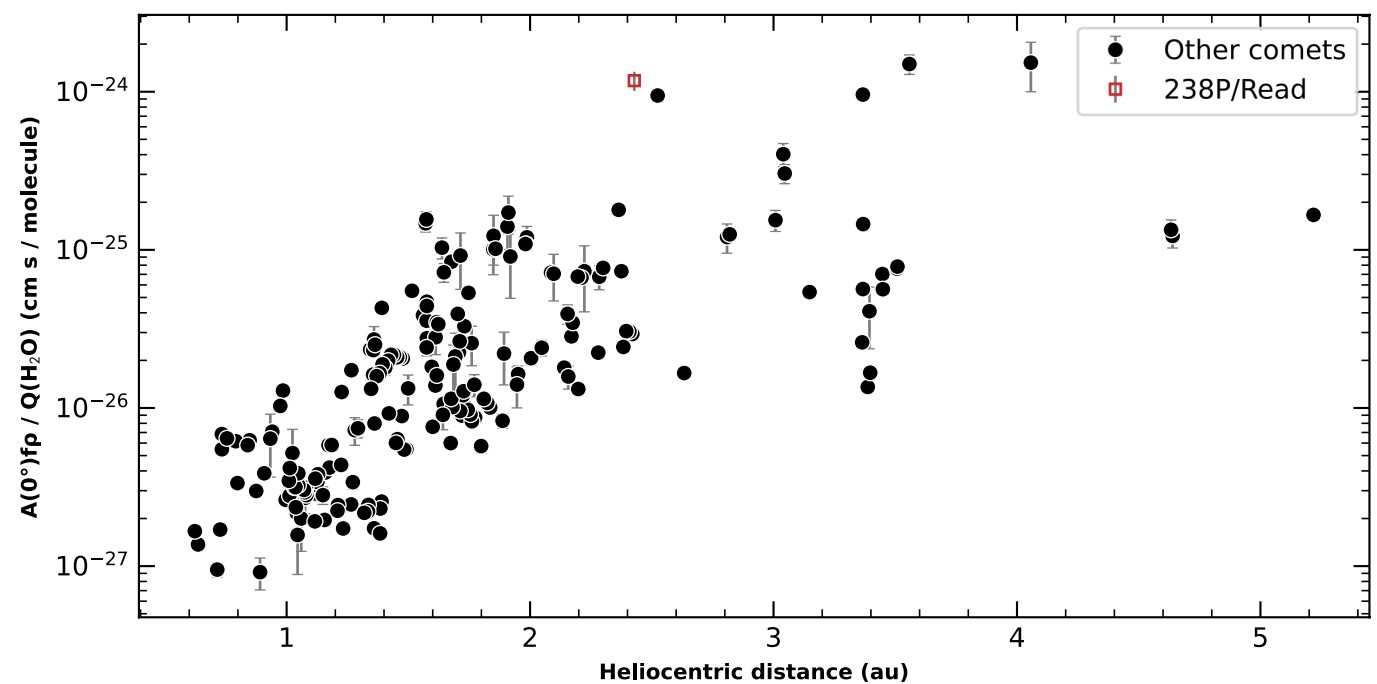

**Extended Data Fig. 5 | Coma dust-to-water ratio for comet 238P/Read and the general comet population.** Error bars represent 1 s.d. The dust content is expressed as the cometary *Afρ* quantity, corrected to a phase angle of 0° and in units of centimeters. The water content is the production rate at the nucleus in units of molecules per second. The Read *Afρ* value has been converted from the near-infrared to an optical *R*-band value. Data for other comets are based on the literature[13]. See Methods for details on the conversions.

**Extended Data Table 1 | Active areas and fractions**

| Model | Species | Production rate ($10^{24}$ molec. s$^{-1}$) | Sublimation rate ($10^{16}$ molec. s$^{-1}$ cm$^{-2}$) | Active area (km$^2$) | Active fraction (%) | Radius circular area (m) |
|---|---|---|---|---|---|---|
| Slow rotator | $H_2O$ | 9.9±0.1 | 3.40 | 0.03 | 4.1 | 97 |
| Rapid rotator | $H_2O$ | 9.9±0.1 | 0.94 | 0.11 | 14.8 | 198 |
| Rapid rotator | $CO_2$ | <0.07 | <12 | $<6\times10^{-5}$ | <0.008 | <4 |

Uncertainties are 1 s.d. There is an additional 10% calibration uncertainty not accounted for in the error bars. The active fraction calculation assumes a 0.24-km radius nucleus, and the radius is that of a circle with an area equal to the active area.