## [Peer Review File · Nature]

Manuscript Title: Spectroscopic identification of water emission from a main-belt comet

Reviewer Comments & Author Rebuttals

Reviewer Reports on the Initial Version:

Referees' comments:

Referee #1 (Remarks to the Author):

- A. The authors have detected water molecules in main-belt asteroid 238P/Read.
- B. Previous work has shown that the properties of 238P can only be explained by the sublimation of water ice. This paper confirms previous work with the first direct detection of gas.
- C. Data and methodology all valid and standard.
- D. Appropriate.
- E. The main conclusion (gas drives the activity) is solid.
- F. Paper is ready to go as it is.
- G. References ok.
- H. This is a very simple paper reliant on a single very convincing measurement to confirm beyond reasonable doubt the nature of activity in 238P.

The paper is appropriate for publication in Nature.

David Jewitt

Referee #2 (Remarks to the Author):

This letter presents the first direct detection of outgassing from a Main Belt Comet, a very important result in the field and one that has been searched for for some time. It is good to see that JWST has its (predicted) capability to make this detection, which opens up a new area in the study of comet activity drivers in weakly active and/or distant comets. The detection of water outgassing is clear and unambiguous, and the strong upper limits on other species are also interesting, showing that this comet (and presumably other MBCs) are different from other short period comets, which has strong implications on solar system evolution models. The paper is well written and presents all results clearly - figures are necessary and clear. I strongly recommend that it is published in Nature, and have only minor comments that the authors should address in a revised version.

Fig 1: The ice features in the 103P spectrum shown here are very hard to see on this scale, and I don't think that these are relevant. It is perhaps confusing to show two different 103P spectra - one would be sufficient to make the comparisons with 238P.

Pg 6. It is perhaps worth noting that MBCs being different to other comets, and therefore the limits on water production derived from CN limits not necessarily being all that strong, was discussed/predicted previously (e.g. in the review by Snodgrass et al that you cite earlier in the paragraph). It is good to have an observational confirmation of this.

Pg 7. The authors give radii for active 'patches' corresponding to the active areas required (in a table in the methods section). The authors could consider giving these radii in the text here, and discussing whether or not these patch sizes support (or not) popular MBC activation theories, such as a recently uncovered area of ice in a crater. At 100-200m in radius the areas are perhaps larger than have been predicted in some models, and would be a large crater, but perhaps not

unexpected on a body this size. Can anything be said about expected impact rates / crater sizes / the population of currently active MBCs that would be expected to correspond to this?

Pg 9/10. The only point in this letter that I have some (small) doubts about is the section on comparing dust-to-gas ratios with other comets. I think that the opening sentence of this paragraph (that dust-to-gas infers 'composition') promises a bit much. There are many unknowns. I think that it is reasonable to say that dust-to-gas can be compared with other comets to say whether or not this MBC is typical or unusual, and that the arguments given show that it could have a range of values that overlap with other comets, but care must be taken not to read too much into it. It would also be worth adding a caveat that the comparison with Rosetta results on 67P is really comparing very different types of measurement (to an already highly debated number).

Pg 22. I'm not sure how helpful the activity timescale calculation is, given the many uncertainties involved, and because this ends with a slightly hand-waving 'few orbits' to end up with a number that is consistent with what we already know observationally - that this (and other) MBCs are active for at least a few orbits. Perhaps this section would be better with a comparison with previous thermal model predictions for active lifetimes for MBCs, and some discussion as to why the initial ~ 1 year timescale is very short.

Fig 4. I assume that the apparent separate bright spot that is just separated from the central condensation in the F227W image is some artefact? It would perhaps be worth commenting on in the caption at least.

Fig 5. The two models shown in this figure are not described in the text. Given that the 2nd model doesn't appear to be a very good fit, and it isn't described anywhere, I don't think it needs to be shown.

Fig 8. Please add an error bar for 238P in this plot (or explain in the caption if this would be too small or too large to plot).

Author Rebuttals to Initial Comments:

ms 2023-03-04464

Kelley et al.

Comments from the reviewers are marked with ">".

We have realized that the sublimation temperatures of ammonia ice and carbon dioxide ice are similar, which adds interesting complexity to the lack of CO₂ coma and the presence of the rounded 3- μ m band, possibly due to ammonium salts. We have added a comment to this effect: "However, ammonia and CO₂ ice have similar sublimation temperatures, therefore it may be that Read had these volatiles in the past, but they have since been lost."

> Fig 1: The ice features in the 103P spectrum shown here are very hard to see on this scale, and I don't think that these are relevant. It is perhaps confusing to show two different 103P spectra - one would be sufficient to make the comparisons with 238P.

This is a good point, and we have removed one, specifically leaving the "icy" spectrum since we do compare to it in Fig. 3. The caption is edited accordingly.

> Pg 6. It is perhaps worth noting that MBCs being different to other comets, and therefore the limits on water production derived from CN limits not necessarily being all that strong, was discussed/predicted previously (e.g. in the review by Snodgrass et al that you cite earlier in the paragraph). It is good to have an observational confirmation of this.

We have added a sentence to explicitly cite this point: "This conclusion is in agreement with previous predictions that the CN/H₂O ratio of the general comet population may not be representative of main-belt comets.[^][Snodgrass et al. 2017]"

> Pg 7. The authors give radii for active 'patches' corresponding to the active areas required (in a table in the methods section). The authors could consider giving these radii in the text here, and discussing whether or not these patch sizes support (or not) popular MBC activation theories, such

as a recently uncovered area of ice in a crater. At 100-200m in radius the areas are perhaps larger than have been predicted in some models, and would be a large crater, but perhaps not unexpected on a body this size. Can anything be said about expected impact rates / crater sizes / the population of currently active MBCs that would be expected to correspond to this?

We have investigated this somewhat, and find that 10-m impactor is needed to account for the active area. However, the energy of such an impact is enough to disrupt the nucleus, given nominal assumptions. Therefore, it initially seems unlikely, but perhaps the parameters of sub-catastrophic impacts may be tuned to produce the needed active area. We state these conclusions in a new paragraph in the main text and a new section in the Methods (pages 3 and 11), but further investigation should be the topic of a separate work.

> Pg 9/10. The only point in this letter that I have some (small) doubts about is the section on comparing dust-to-gas ratios with other comets. I think that the opening sentence of this paragraph (that dust-to-gas infers 'composition') promises a bit much. There are many unknowns. I think that it is reasonable to say that dust-to-gas can be compared with other comets to say whether or not this MBC is typical or unusual, and that the arguments given show that it could have a range of values that overlap with other comets, but care must be taken not to read too much into it. It would also be worth adding a caveat that the comparison with Rosetta results on 67P is really comparing very different types of measurement (to an already highly debated number).

Certainly the method by which one would infer the interior properties based on the coma demands assumptions, which we largely left unstated. Regardless, we agree with the reviewer that there are many unknowns. This paragraph, now in the Methods, as been revised. The intro paragraph to the sub-section "Dust-to-gas ratio" now reads: "The coma dust-to-gas ratio may be measured from our data and compared to other comets." We quote ~ 1 instead of 0.85 for the dust-to-volatiles mass ratio of 67P, and refer the reader to Choukroun et al. (2020) for more discussion on the topic. Finally, our concluding sentence has been revised accordingly: "Therefore, our conclusions are that comet Read has a [coma] dust-to-gas ratio [broadly] consistent with the general comet population, which [suggests it may have] formed in a region of the protoplanetary disk with abundant water ice."

> Pg 22. I'm not sure how helpful the activity timescale calculation is, given the many uncertainties involved, and because this ends with a slightly hand-waving 'few orbits' to end up with a number that is consistent with what we already know observationally - that this (and other) MBCs are active for at least a few orbits. Perhaps this section would be better with a comparison with previous

thermal model predictions for active lifetimes for MBCs, and some discussion as to why the initial ~ 1 year timescale is very short.

We now reference the more physical analyses of Prialnik and Rosenberg (2009) and Schorghofer (2016). But since this is the first water production rate measurement, we think that some attempt to use the observed value is useful. To avoid the hand waviness, we have also revised our calculation to instead estimate the total water mass lost considering the rh^{*-3} activity dependence from the model of Hsieh et al. (2009). In our revised text, the dust-to-gas mass loss rate ratio leads us to suggest that the ice layer retreats faster than the surface. Therefore, the surface devolatilizes with time, which would ultimately quench activity.

> Fig 4. I assume that the apparent separate bright spot that is just separated from the central condensation in the F227W image is some artefact? It would perhaps be worth commenting on in the caption at least.

Indeed, it appears to be an artifact in a single frame, perhaps due to $1/f$ noise on the detector. We have commented on this fact in the caption.

> Fig 5. The two models shown in this figure are not described in the text. Given that the 2nd model doesn't appear to be a very good fit, and it isn't described anywhere, I don't think it needs to be shown.

The second model was a fit to a smaller wavelength range and we have removed it from the figure. In the "Methods: Reflectance spectrum sub-section" we clarify that two models with different wavelength ranges were fit to the data and analyzed to assess the thermal emission.

> Fig 8. Please add an error bar for 238P in this plot (or explain in the caption if this would be too small or too large to plot).

The error bar was about the same size as the marker. However, we have revised our uncertainty calculation on this value. Because this ratio is based on a comparison of NIRSpec and NIRCам data,

the absolute calibration uncertainty needs to be accounted for. The log10 uncertainty increased from 0.04 to 0.06. We have also made the marker fill color transparent so that the error bar can be more clearly seen.